# G-protein coupled estrogen receptor 1, amyloid-β, and tau tangles in older adults

Shahram Oveisgharan [1,2] ✉, Lei Yu [1,2], Katia de Paiva Lopes[1,2], Vladislav A. Petyuk [3], Shinya Tasaki[1,2], Ricardo Vialle [1,2], Vilas Menon [4], Yanling Wang[1], Philip L. De Jager [5,6], Julie A. Schneider[1,2,7] & David A. Bennett[1,2]

Accumulation of amyloid-β (Aβ) and tau tangles are hallmarks of Alzheimer's disease. Aβ is extracellular while tau tangles are typically intracellular, and it is unknown how these two proteinopathies are connected. Here, we use data of 1206 elders and test that RNA expression levels of *GPER1*, a transmembrane protein, modify the association of Aβ with tau tangles. *GPER1* RNA expression is related to more tau tangles ($p = 0.001$). Moreover, *GPER1* expression modifies the association of immunohistochemistry-derived Aβ load with tau tangles ($p = 0.044$). Similarly, *GPER1* expression modifies the association between Aβ proteoforms and tau tangles: total Aβ protein ($p = 0.030$) and Aβ38 peptide ($p = 0.002$). Using single nuclei RNA-seq indicates that *GPER1* RNA expression in astrocytes modifies the relation of Aβ load with tau tangles ($p = 0.002$), but not *GPER1* in excitatory neurons or endothelial cells. We conclude that *GPER1* may be a link between Aβ and tau tangles driven mainly by astrocytic *GPER1* expression.

Accumulation of amyloid-β (Aβ) and tau tangles are hallmarks of Alzheimer's disease (AD)[1]. It is hypothesized that Aβ accelerates tau phosphorylation and tau tangle formation leading to dementia. However, Aβ is extracellular while tau tangles are typically intracellular[2], and a large body of work is ongoing to address how these two proteinopathies interact to cause AD[3]. This line of work is important because of epidemiological studies that reported faster cognitive decline when both Aβ and tau present in the brain, compared with the presence of either one alone[4], and because of null or small effect clinical trials that targeted removal of either Aβ or tau deposits from the brain[5,6]. Identification of proteins and pathways that connect Aβ and tau tangles may provide druggable targets to interfere with the progression of AD and subsequent cognitive decline.

Several mechanisms have been suggested for linking Aβ with tau tangles[3]. Hyperphosphorylation of tau protein is a crucial step causing dissociation of tau protein from neuronal microtubules and formation of tau tangles. Multiple kinases including cyclin-dependent kinase-5 and glycogen synthase kinase-3β may phosphorylate tau at multiple sites, and studies reported activation of these kinases by Aβ[7]. Other suggested mechanisms for linking Aβ with tau tangles are Aβ-induced activation of caspase-3 and

production of tau fragments that easily assemble to tau fibrils[8], promotion of Fyn kinase phosphorylation that together with phosphorylated tau binds to N-methyl1-1-D-aspartate (NMDA) receptor and activate excitotoxic signaling mechanism[9], and mitochondrial damage and increased reactive oxygen species because of intracellular Aβ and phosphorylated tau. Studies have suggested that Aβ oligomers including an amino-terminal truncated form[10] are more toxic than extracellular Aβ plaques as the oligomers can possibly be transported inside neurons through Aβ receptors[11] and interact with tau protein.

Women have higher levels of tau tangles compared with men[12,13], and low estrogen levels in postmenopausal women have been suggested to contribute to vulnerability of women to AD dementia[14,15]. To address a possible role for estrogen in more tau in women's brains[16], in a prior study we examined estrogen receptors, estrogen receptor α (ER1) and β (ER2) and the transmembrane receptor G protein-coupled estrogen receptor (GPER1), in the association with tau tangles and cognitive decline[17]. We found that a greater RNA expression of *GPER1*, not *ER1* or *ER2*, was associated with more tau tangles and faster cognitive decline in women[17]. These findings and the transmembrane location of GPER1 were the basis of

[1]Rush Alzheimer's Disease Center, Rush University Medical Center, Chicago, IL, USA. [2]Department of Neurological Sciences, Rush University Medical Center, Chicago, IL, USA. [3]Biological Sciences Division, Pacific Northwest National Laboratory, Richland, WA, USA. [4]Department of Neurology and Taub Institute for Research on Alzheimer's Disease and the Aging Brain, Center for Translational and Computational Neuroimmunology, Columbia University Irving Medical Center, New York, NY, USA. [5]Department of Neurology, Center for Translational and Computational Neuroimmunology, Columbia University Irving Medical Center, New York, NY, USA. [6]Taub Institute for Research on Alzheimer's Disease and the Aging Brain, Columbia University Irving Medical Center, New York, NY, USA. [7]Department of Pathology, Rush University Medical Center, Chicago, IL, USA. ✉e-mail: shahram_oveisgharan@rush.edu

our hypothesis that GPER1 might contribute to a pathway that connects extracellular Aβ to intracellular tau tangles. To test this hypothesis, we extended our prior work in several ways. First, we examined an interaction between Aβ and *GPER1* in relation to tau as we expected to observe a stronger association between Aβ and tau tangles among persons with higher levels of *GPER1*. Second, we examined both immunohistochemical and proteomic measures of Aβ in relation to *GPER1* and tau tangles. Third, we examined multiple signaling mechanisms of GPER1 to test whether they are also involved in the pathway linking Aβ with tau tangles. Fourth, we examined single nuclei RNA expression (snRNA-seq) data to identify whether specific cell types' *GPER1* underlie *GPER1* relation with Aβ and tau tangles.

## Results

Characteristics of participants are summarized in Table 1. Participants were on average 89.6 (SD = 6.6) years old at the time of death, two thirds were women, and 42% (*n* = 507) had AD dementia. In postmortem examination, 65% (*n* = 775) had AD pathological diagnosis. Higher stages of Aβ deposition, indicating distribution of Aβ from neocortex to basal ganglia and brain stem, was associated with more advanced stages of tau tangle deposits where tau tangles extend from mesial temporal to neocortex (Spearman $\rho = 0.60$, $p < 0.001$).

### GPER1 and Aβ

In a linear regression model controlled for age at death, sex, and education, higher levels of *GPER1* RNA expression and Aβ load were associated with higher tau tangle density (Table 2-*Series A*). When we added an interaction term between *GPER1* RNA expression level and Aβ load, the interaction term was significant, indicating that *GPER1* RNA expression level modified the association between Aβ and tau tangles (Fig. 1). Calculation of the effect size indicated that per one unit more Aβ load, tau tangle density was 14% more when *GPER1* expression level was at the 90th percentile compared to the 10th percentile (Fig. 1). We also examined associations of Aβ load with tau tangle density in the 3 tertiles of *GPER1* RNA expression. The estimates of the associations between Aβ load and tau tangle density were 12.5% and 36.1% stronger in the second (estimate=0.495, SE = 0.047, $p < 0.001$) and third (estimate = 0.599, SE = 0.055, $p < 0.001$) tertiles of *GPER1* RNA expression compared with the first tertile (estimate = 0.440, SE = 0.046, $p < 0.001$).

Next, we replaced Aβ load, an immunohistochemical measure of Aβ plaques, with cortical Aβ protein level assessed by targeted proteomics analysis, which also captures soluble Aβ among persons without deposited Aβ[18]. Two Aβ proteoforms were quantified, total Aβ and Aβ38. In 2 separate linear regression models, higher levels of both Aβ proteoforms (Table 2-*Series B and C*) were associated with greater tau tangle density. Moreover, *GPER1* RNA expression level modified the associations of both proteoforms with tau tangles (Fig. 1).

Estrogen level is lower in postmenopausal women compared with men[19], which might change the interaction between *GPER1* RNA expression and Aβ across sexes. In the current analyses, *GPER1* RNA expression level was higher in women compared with men (13.88 (SD = 0.93) vs. 13.71 (SD = 0.84), $p = 0.001$), an indirect measure of different estrogen levels in the 2 sexes. Therefore, we estimated the interactions between *GPER1* RNA expression and Aβ load in the association with tau tangles in men and

## Table 1 | Characteristics of study participants (*n* = 1206)

| Characteristics | Mean (SD) or *n* (%) |
|---|---|
| Age at death baseline, years, Mean (SD) | 89.6 (6.6) |
| Women, *n* (%) | 820 (68.0) |
| Education, years, Mean (SD) | 16.2 (3.5) |
| White non-Hispanic, *n* (%) | 1167 (96.8) |
| Mini Mental State Examination score, Mean (SD) | 20.8 (9.2) |
| Alzheimer's dementia, *n* (%) | 507 (42.0) |
| Estrogen-related medications | |
| Estrogens, *n* (%) | 91 (7.6) |
| Selective estrogen receptor modulators, *n* (%) | 72 (6.0) |
| Aromatase inhibitors, *n* (%) | 23 (1.9) |
| Pathological diagnosis of Alzheimer's disease, *n* (%) | 775 (64.3) |
| Modified Thal Amyloid-β score | |
| A0, *n* (%) | 201 (16.7) |
| A1, *n* (%) | 206 (17.1) |
| A2, *n* (%) | 348 (28.9) |
| A3, *n* (%) | 451 (37.4) |
| Braak stages of neurofibrillary tangles | |
| 0–II, *n* (%) | 201 (16.7) |
| III, *n* (%) | 300 (24.9) |
| IV, *n* (%) | 396 (32.8) |
| V–VI, *n* (%) | 309 (25.6) |
| Global burden of amyloid-β and tau tangles | |
| Square root of amyloid-β load, Mean (SD) | 1.6 (1.2) |
| Square root of tau tangle density, Mean (SD) | 1.6 (1.3) |

## Table 2 | Associations of different measures of amyloid-β (Aβ) with tau tangles and modifications of the associations by *GPER1* RNA expression level

| Series | Model terms | Model 1 Estimate (SE), *p* value | Model 2 |
|---|---|---|---|
| A | *GPER1* RNA expression level | 0.118 (0.036), 0.001 | 0.008 (0.066), 0.909 |
| | Aβ load | 0.523 (0.029), <0.001 | 0.533 (0.029), <0.001 |
| | *GPER1*×Aβ load | NA | 0.063 (0.032), 0.044 |
| B | *GPER1* RNA expression level | 0.128 (0.040), 0.001 | 0.166 (0.043), <0.001 |
| | Total Aβ protein level | 0.178 (0.012), <0.001 | 0.182 (0.012), <0.001 |
| | *GPER1*×total Aβ | NA | 0.031 (0.014), 0.030 |
| C | *GPER1* RNA expression level | 0.121 (0.043), 0.005 | 0.102 (0.043), 0.018 |
| | Aβ38 peptide level | 0.154 (0.029), <0.001 | 0.169 (0.030), <0.001 |
| | *GPER1*×Aβ38 | NA | 0.103 (0.033), 0.002 |

In three series of linear regressions, associations of *GPER1* RNA expression level and three measures of Aβ were examined with tau tangles density (as the outcome). The three measures of Aβ were Aβ load, determined using immunohistochemical methods, and total Aβ and Aβ38 protein levels, determined by targeted proteomic analysis. In each series of linear regressions one of the Aβ measures was examined in two models. Model 1 included terms for Aβ measure and *GPER1* RNA expression level, and model 2 included model 1 terms and a term for an interaction between the Aβ measure and *GPER1* RNA expression level. All the models were controlled for age at death, sex, and education.

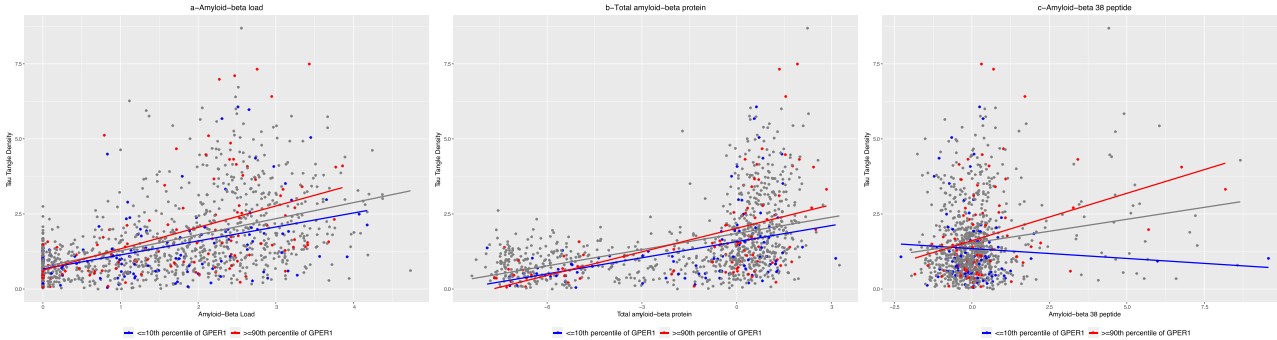

**Fig. 1 | Modification of the associations between Aβ load, total Aβ protein, and Aβ38 peptide with tau tangles density by *GPER1* RNA expression level.** The figure has 3 panels illustrating scatter plots of the associations between immunohistochemistry-derived Aβ load (**a**), total Aβ protein (**b**), or Aβ38 peptide (**c**) and tau tangle density. The dots and the regression line in each panel are colored red if the corresponding *GPER1* RNA expression level is high (≥90th percentile) or are colored blue if the corresponding *GPER1* RNA expression is low (≤10th percentile). The panels include 1198 participants when examining Aβ load, and 1001 and 1003 participants when examining, total Aβ protein, or Aβ38 peptide, respectively.

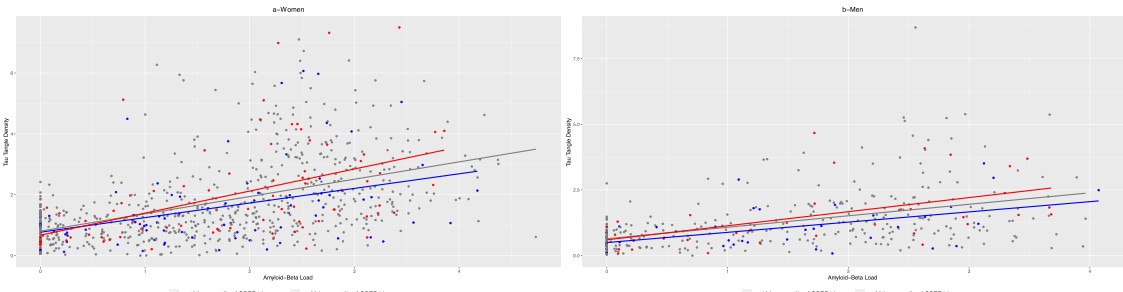

**Fig. 2 | Modification of the association between Aβ load and tau tangles density by *GPER1* RNA expression level in women and men.** The figure has 2 panels illustrating scatter plots of the associations between Aβ load and tau tangle density in women (**a**) and men (**b**). The dots and the regression line in each panel are colored red if the corresponding *GPER1* RNA expression level is high (≥90th percentile of sex-specific *GPER1* level) or are colored blue if the corresponding *GPER1* RNA expression is low (≤10th percentile of sex-specific *GPER1* level). The plot-a includes women (*n* = 814) and the plot-b includes men (*n* = 384).

women separately (Fig. 2). The interaction estimates were similar between women and men [(women: estimate=0.058, SE = 0.039, *p* = 0.137); (men: estimate=0.057, SE = 0.054, *p* = 0.296)]. These findings suggested that *GPER1* RNA expression level modified the association between Aβ and tau tangles in men and women. Furthermore, controlling the main models for estrogen-related medications did not change the modification of the association between Aβ and tau tangles by *GPER1* (Supplementary Table 1).

In subsets of participants *GPER1* RNA expression levels were also determined in the posterior cingulate cortex (*n* = 633) and the anterior caudate (*n* = 687). Although *GPER1* levels in the 3 brain regions were different (Supplementary Table 2), the levels were moderately correlated (all 3 ρ > 0.5, Supplementary Table 2). Like *GPER1* level in DLPFC, *GPER1* RNA level in the posterior cingulate cortex was related to higher levels of tau tangles and modified the association of Aβ load with tau tangles, but not *GPER1* in the anterior caudate (Supplementary Table 3). When we replaced global measures of Aβ load and tau tangle density with the stages of Aβ and tau tangle distribution across the brain, the main findings persisted. A higher *GPER1* RNA level in DLPFC was associated with more advanced stages of tau tangles (Supplementary Table 4). In addition, *GPER1* RNA level in both DLPFC and the posterior cingulate cortex had a borderline interaction with the stage of Aβ in relation to tau tangle stages. However, *GPER1* RNA level in the anterior caudate did not have these associations (Supplementary Table 4). *GPER1* RNA level in neither of the brain regions was associated with the stage of Aβ or Aβ load (Supplementary Table 5).

## GPER1 signaling mechanisms and Aβ
Activation of GPER1, which is a transmembrane receptor, results in activation of signaling mechanisms including c-Jun N-terminal kinase (JNK), extracellular signal-regulated kinase (ERK), Akt kinase[20], adenylate cyclase,

protein kinase A[21], and Phospholipase C Beta (PLCβ)[22]. We tested whether there was an interaction between RNA expressions and protein levels of these signaling mechanisms and Aβ in relation to tau tangles.

We first examined the correlations between RNA expression levels of *GPER1* and of the genes of downstream signaling mechanisms indicating some coordinated expressions (Supplementary Table 6). Then, we examined whether RNA expressions of GPER1 signaling mechanisms modified the associations of Aβ load with tau tangles. In linear regression models that included terms for RNA expression of each gene separately, Aβ load, and interaction of the gene with Aβ, only RNA expression of *PLCβ 1* had an interaction with Aβ load (Supplementary Table 7).

Next, we examined protein levels of GPER1 signaling mechanisms. *GPER1* RNA expression level was correlated with protein levels of some of the signaling mechanisms (Supplementary Table 6). When we examined the protein levels of GPER1 signaling mechanisms to find if they modified the associations between Aβ load with tau tangles, we found that none of the proteins, even PLCβ 1, had the modification effect (Supplementary Table 7).

Because several of the examined genes were related to tau phosphorylation and tau tangles in prior studies, we examined the GPER1 signaling mechanisms in relation to tau tangles. From the 28 examined RNAs, expression levels of 12 were related to tau tangles. However, only RNA level of one of them remained significantly related to tau tangles when the model also included *GPER1*, Aβ load, and their interaction terms (Supplementary Table 8). Similarly, from the 26 proteins of GPER1 signaling mechanisms we found 11 to be related to tau tangles with their associations being attenuated in the presence of *GPER1*, Aβ, and their interaction (Supplementary Table 8). These findings suggest that some of the examined proteins are not directly related to tau tangles, but rather are associated by a complex relationship that involves *GPER1*, Aβ, and their interaction.

## GPER1, Aβ, and autophagy pathway

Autophagy is a cellular process characterized by formation of double-membrane vesicles (autophagosomes) around cytoplasmic organelles and molecules, which will coalesce with lysosomes for degradation of engulfed materials and recycle of basic nutrimental needs. Autophagy acts during cellular stress including nutrient deprivation and for degrading damaged organelles and proteins[23].

Prior studies reported involvement of estrogen and estrogen receptors in autophagy[24] and autophagy-related tau clearance[25]. Therefore, we examined RNA expression and protein levels of autophagy genes in relation to *GPER1* RNA level and tau tangles. *GPER1* RNA expression was related to levels of some of the autophagy genes' RNA expressions and proteins (Supplementary Table 9). However, none of the 22 autophagy genes' RNA expression and only 2 of the 18 protein levels modified the association between Aβ load and tau tangles (Supplementary Table 10). In addition, while some of the autophagy genes' RNA expressions and protein levels were associated with tau tangles their associations were attenuated in the presence of *GPER1*, Aβ, and their interaction (Supplementary Table 11). These findings suggest that a complex relationship exists between autophagy, Aβ, tau tangles, and related molecular pathways.

## GPER1 comembers in a class of G-protein-coupled receptors

GPER1 is a member of the superfamily of G-protein-coupled receptors that in humans includes more than 800 proteins with a common feature of having 7 transmembrane helices[26,27]. Although most of the G-protein-coupled receptors are classified into 5 major categories, GPER1 is one of the 23 proteins that is not a member of one of these major categories and is under a separate category of G-protein-coupled receptors[26]. We examined whether an interaction was also observed with RNA expression of the 22 other proteins co-classified with GPER1. RNA expressions of 14 of the 22

genes had passed quality control criteria. However, none of the genes had an interaction with Aβ load in the association with tau tangles (Supplementary Table 12). This analysis suggested that the interaction was specific to *GPER1*.

## snRNA-seq expressions of *GPER1*

To identify whether the associations obtained by examining bulk RNA expression were due to specific cell types, we used snRNA-seq data to examine *GPER1* RNA expression levels in different cells. snRNA-seq data were available in 419 participants who were similar to participants without snRNA-seq data in age at death, sex, frequency of AD pathological diagnosis, and Aβ load, but had fewer tau tangles (Supplementary Table 13).

We found that *GPER1* RNA expression pass quality control criteria only in astrocytes, excitatory neurons, and endothelial cells, but not inhibitory neurons, microglia, oligodendroglia, and oligodendrocyte precursor cells (Fig. 3). *GPER1* RNA expression levels in the astrocytes and excitatory neurons were correlated (Spearman $\rho = 0.25$, $p < 0.001$), but they were not correlated with that of endothelial cells (Supplementary Table 14).

Here, higher levels of *GPER1* in the astrocytes were associated with more tau tangles (estimate=0.219, SE = 0.072, $p = 0.003$). However, this association was not observed between excitatory neurons and endothelial cells *GPER1* RNA expression and tau tangles (excitatory neurons: estimate=0.045, SE = 0.084, $p = 0.594$; endothelial cells: estimate=0.067, SE = 0.050, $p = 0.181$). Moreover, astrocytes *GPER1* RNA expression modified the association between Aβ load and tau tangles (Astrocytes GPER1×Aβ: estimate=0.174, SE = 0.056, $p = 0.002$). Calculation of the effect size indicated that for each unit higher Aβ load, tau tangle density was higher by 116% when astrocytes GPER1 level was at the 90th percentile compared with the 10th percentile (Fig. 4). By contrast, we did not observe a significant interaction between excitatory neurons or endothelial cells *GPER1* RNA expression and Aβ in the association with tau tangles (excitatory neurons:

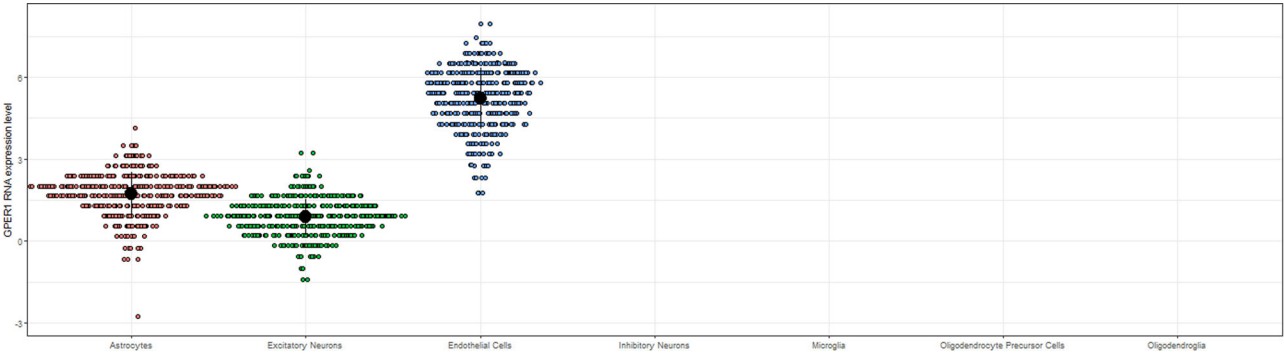

**Fig. 3 | *GPER1* RNA expression levels in different cell types.** Figure 3 is derived from analyzing participants with single nuclei RNA expression data ($n = 419$).

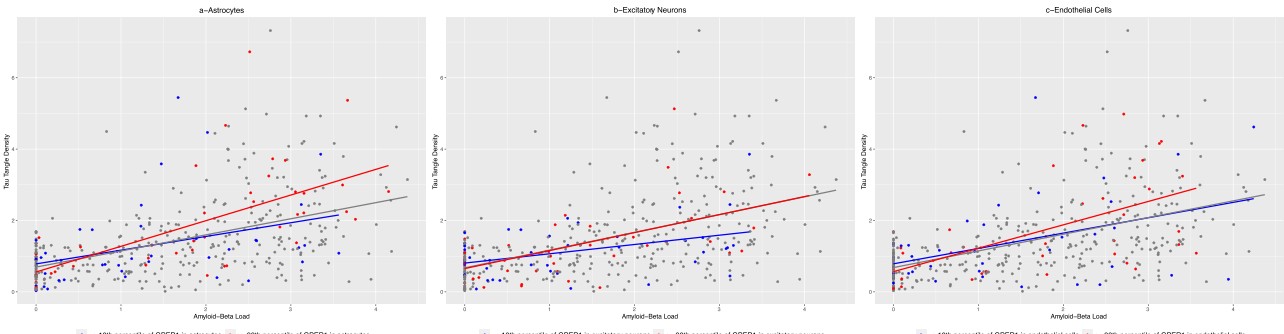

**Fig. 4 | Modification of the associations between Aβ and tau tangles by astrocytic, excitatory neurons, and endothelial cells *GPER1* RNA expression levels.** The figure has 3 panels illustrating scatter plots of the associations between Aβ load and tau tangle density delineating participants by their *GPER1* RNA expression level in the astrocytes (**a**), excitatory neurons (**b**), and endothelial cells (**c**). The dots and the

regression line in each panel are colored red if the corresponding single nuclei RNA expression level of *GPER1* is high (≥90th percentile) or are colored blue if the corresponding single nuclei RNA expression of *GPER1* is low (≤10th percentile). Each panel includes 417 participants with Aβ load, tau tangle density, and single nuclei RNA expression data.

estimate=0.110, SE = 0.066, $p$ = 0.094; endothelial cells: estimate=0.008, SE = 0.035, $p$ = 0.828).

## Astrocytic cytokines

Astrocytes are a key player in neuroinflammation, which is related to AD[28]. Therefore, we examined RNA expressions of 47 inflammatory cytokines using snRNA-seq data and found 12 of them expressed by astrocytes (Supplementary Fig. 1). *GPER1* RNA expression had a null to weak correlation with the RNA expressions of the 12 astrocytic cytokines (Supplementary Table 15). However, RNA expressions of none of the astrocytic cytokines modified the association between Aβ and tau (Supplementary Table 16). Moreover, while RNA expressions of 4 of the cytokines were related to tau tangles, their associations were attenuated and not significant after inclusion of Aβ, *GPER1* RNA expression, and their interaction (Supplementary Table 17). The above analyses indicated that the pathway connecting GPER1 and Aβ with tau does not involve astrocytic cytokines.

## Discussion

Using postmortem pathological, bulk tissue and single nuclei RNA expression, and proteomic data from more than 1200 community-dwelling older adults, we found that *GPER1* modified the association of Aβ with tau tangles as the association between Aβ with tau tangles was stronger at higher levels of *GPER1* RNA expression. This association was specific to *GPER1*; RNA expressions of genes of signaling mechanisms of GPER1 or GPER1 co-classified G-proteins coupled receptors did not modify the association between Aβ and tau tangles. Examining snRNA-seq data revealed that it was astrocytic *GPER1* RNA expression that modified the association between Aβ and tau. These findings suggest that astrocytic GPER1 may be involved in a pathway that links extracellular Aβ and intracellular tau.

Prior studies suggest that accumulation of Aβ initiates the AD cascade[29], which accelerates accumulation of tau tangles, the main cognitive declining factor in AD[30,31]. Null or modest benefits and side effects of immunotherapies that target Aβ[32] removal suggest that these agents may be less effective among persons with tau tangles that correlate much more strongly with cognitive decline[33]. Therefore, the field is changing gear investigating mechanisms that connect Aβ with tau to find agents that interfere with Aβ-induced tau tangle formation, which may be a more promising therapeutic target. However, most prior studies examining molecular mechanisms linking Aβ-tau were experimental, using cell lines, brain organelles, and animal models, limiting generalizing their findings to humans[34]. Our study extends prior studies as it examined RNA expression of *GPER1*, derived from bulk brain tissue and single nuclei, brain pathologies, and proteins from a large number of community-dwelling older adults, and found that *GPER1* modified the relationship between Aβ and tau.

Different mechanisms have been suggested to link Aβ with tau including tau aggregation directly triggered by Aβ[35], activation of microglia[36], hyperexcitation of neurons induced by Aβ[37], and cerebral hypoperfusion[38]. Our finding that astrocytic *GPER1* modified the association between Aβ and tau tangles is an indirect evidence that GPER1 may be a link between Aβ and tau. Astrocytes were a key mediator in Aβ-induced tau phosphorylation in neuronal cultures[39], and in linking participant's Aβ burden to plasma phosphorylated tau level[40]. Our analysis did not support release of inflammatory cytokines by activated astrocytes as a mechanism linking Aβ and tau tangles through astrocytic GPER1. One possible mechanism is GPER1-mediated internalizing of Aβ in the astrocytes[41,42] that triggers formation of astrocytic tau aggregates[43], which act as seeds for formation of neuronal tau tangles. This hypothesis is supported by prior studies that showed presence of Aβ oligomers inside the cells and induction of tau phosphorylation after intraventricular injection of Aβ oligomers in monkeys[44] or treating cell cultures by solutions containing soluble Aβ oligomers[45].

ER1 and ER2 are essentially nuclear estrogen receptors that after activation by estrogen bind transcription factors and regulate transcription of a variety of genes involved in different cellular processes. Besides the genomic effect that occurs in the time frame of hours to days, estrogen has an instant non-genomic effect that starts in the cells within minutes of estrogen exposure, contributes to estrogen-induced neuroprotection[46] and cognitive function[47], and is mediated by estrogen surface receptors including GPER1[48]. In fact, prior experimental studies reported that activation of GPER1 by estrogen has cell-type specific consequences as it promotes cell survival in neurons and causes apoptosis in astrocytes[49]. These prior studies together with our finding about modification of the relationship between Aβ and tau tangles by astrocytic *GPER1* suggest a mechanism for more tau tangles[16] and higher prevalence of Alzheimer's dementia observed in women[50]. Postmenopausal women have low estrogen levels and high GPER1 that contribute to aggravated Aβ-induced tau hyperphosphorylation and deposition. However, in our study we did not measure estrogen level in DLPFC tissue. Because local estrogen is synthesized in the brain[51] that may offset low levels of estrogen in the circulation, further studies are needed to explicate the relationship between estrogen, GPER1, Aβ, and tau in women and men.

*GPER1* RNA expression level in the anterior caudate was not related to tau tangles and did not modify the associations of Aβ with tau tangles, in contrast to *GPER1* RNA level in DLPFC and the posterior cingulate cortex. Activation of neuronal GPER1 had different effects dependent on the neuronal region[52], which may underlie heterogenous associations of *GPER1* with tau tangles in the current study. However, we did not measure Aβ in the caudate, which might be lower than Aβ level in DLPFC considering distribution pattern of Aβ that begins from neocortex and spreads down to the basal ganglia including caudate nucleus and may underlie lack of association between *GPER1* RNA level in the anterior caudate and tau tangles.

Besides using immunohistochemical methods to measure Aβ load, we also measured level of Aβ through targeted proteomic analysis, which relies on the identification of unique protein sequences. The only two Aβ species[53] quantified were the generic peptide that maps to the middle of the sequence of any Aβ species, identified as total Aβ protein, and a C-terminal fragment of Aβ38. Notably, we were not able to quantify Aβ42 because the derived C-terminal peptide (that is specific to Aβ42) is very hydrophobic that made it incompatible with the current generic sample preparation technique used in mass spectrometry proteomics. However, recent studies find that Aβ38 is elevated in both familial and sporadic forms of AD[54], which support using Aβ38 in our study.

Approximately all GPER1 signaling mechanisms proteins were kinases that act through phosphorylating their target. As abnormal tauopathy is initiated by hyperphosphorylation of tau, one hypothesis was that activation of GPER1 is transduced intracellularly by activation of these kinases that ends in hyperphosphorylation of tau and production of tau tangles. This hypothesis was supported by our finding that the protein levels of several of these kinases were related to tau tangle density. However, the relationship between the kinase levels and tau tangle density was attenuated in the presence of Aβ, GPER1, and their interaction in the model, which is not consistent with the kinases being mediators of the activation of GPER1 by Aβ. In statistical mediation, the association between the mediator and the outcome does not attenuate in the presence of the upstream cause. Therefore, we concluded that a complex relationship exists between Aβ, GPER1 and their signaling mechanisms, and tau tangles, which requires further studies for clarification.

Several strengths support our findings. Multiple streams of data were available, including clinical, pathological, transcriptomic, and proteomic, prepared from 12 hundred community-dwelling older adults who were followed for years and underwent autopsy for brain examination. Personnel who collected pathological or omics data were blind to clinical data of the participants. Availability of single nuclei RNA sequencing data in more than 400 of the participants enabled identification of cells that drive modifying association of Aβ with tau. However, several limitations must be noted. All the findings were derived from an observational study, and our main finding that astrocytic *GPER1* modifies the association between Aβ and tau does not imply a cause and effect finding unless confirmed by future experimental studies. Although many molecular pathways were examined, still some

unmeasured confounders rather than GPER1 may underlie the modification of the relation between Aβ and tau. The study used RNA expression of *GPER1* rather than using GPER1 protein level that is a more genuine measure to be examined in relation to Aβ and tau. However, GPER1 protein level was currently not available in the proteomic analyses of the brain tissues[55,56] and is challenging to measure using conventional LC-MS/MS proteomics protocols due to its hydrophobicity. Single nuclei measurement of *GPER1* RNA expression was not available in all participants to confirm dominant role of astrocytic *GPER1* in modifying the association between Aβ and tau tangles. We tested RNA and protein levels of GPER1 signaling mechanisms including JNK, ERK, and Akt kinases instead of their activation level. The current study examined GPER1 as a transmembrane estrogen receptor with a potential to link extracellular Aβ with intracellular tau tangles. Future studies should examine other receptors of sex and steroid hormones that are located intracellularly and may be involved in AD pathophysiology.

## Methods

### Participants
Participants were from one of two longitudinal ongoing clinical pathological studies of aging and dementia, the Religious Orders Study (ROS) or the Rush Memory and Aging Project (MAP) conducted by Rush Alzheimer's Disease Center. ROS began enrollment in 1994, and eligible candidates were older nuns, priests, and brothers across the United States who were without known dementia and consented to annual clinical evaluations and brain donation at the time of death. MAP began enrollment in 1997, and eligible candidates were older Illinoisans living in retirement facilities or personal accommodations in northeastern Illinois who were without known dementia and consented for clinical evaluations and brain donations. Both studies were approved by an Institutional Review Board of Rush University Medical Center. All participants signed an informed consent and Anatomic Gift Act, and all ethical regulations relevant to human research participants were followed. A large common core of study protocols and data collection methods were implemented in both studies by the same personnel facilitating joint analysis. More details about the studies are provided elsewhere[57].

Of 3734 ROSMAP participants with completed baseline evaluations till April 2022, 1836 had died with completed postmortem pathological assessments. Of the 1836 participants, the first 1206 whose dorsolateral prefrontal (DLPFC) tissues had been examined for transcriptomics comprised the analytic sample. RNA sequencing is ongoing.

### Postmortem pathological assessment
The median postmortem interval was 6.7 (IQR: 5.0–9.3) h. After brain removal, the two hemispheres were separated. One hemisphere was fixed in 4% paraformaldehyde solution for the pathological assessments and the other hemisphere was frozen for future work including transcriptomic and proteomic analyses. The fixed hemisphere was cut into slabs and brain sections were prepared from predetermined brain regions.

### Aβ load
Sections were prepared from multiple brain regions including anterior cingulate cortex, superior frontal cortex, mid frontal cortex, inferior temporal cortex, hippocampus, entorhinal cortex, angular gyrus/supramarginal cortex, and calcarine cortex. Sections (Supplementary Fig. 2) were immunohistochemically stained using antibodies specific for Aβ [6 F/3D (1:50, Dako North America Inc., Carpinteria, CA); 10D5 (1:600, Elan Pharmaceuticals, San Francisco, CA); 4G8 (1:9000, Covance Labs, Madison, WI)]. Digital image analysis was employed to calculate percentages of the sections occupied by immunohistochemically-labeled areas. The averages of the percent areas were calculated to yield regional Aβ loads, which were subsequently averaged to yield brain Aβ load because of being highly correlated (Supplementary Table 18)[16,30]. In addition to the Aβ load, we used a modified 4-level Thal stages to determine distribution of Aβ across the brain. Stage 0 indicates no Aβ, stage 1 indicates presence of Aβ in the neocortex with or without

hippocampus, stage 2 is stage 1 + Aβ in the basal ganglia, and stage 3 is stage 2 + Aβ in the brainstem and cerebellum[2,58].

### Tau tangle density
Sections (Supplementary Fig. 2) from the same brain regions were also immunohistochemically stained using an antibody specific for phosphorylated tau [AT8 (1:1000, Innogenetics, Alpharetta, GA)]. Microscopes equipped with a computer-aided stereology program were used to count tau-labeled neurofibrillary tangles, which were summarized as regional and brain tau tangle density[16,30]. In addition to the tau tangle density, we used a modified Bielschowsky silver stain to visualize neurofibrillary tangles and determine their distribution using Braak stages. Stage 0 indicates no neurofibrillary tangle, stages I and II indicate presence of neurofibrillary tangles in the entorhinal cortex, stages III and IV are stage II + neurofibrillary tangles in the limbic system including hippocampus, and stages V and VI are stage IV + neurofibrillary tangles in the neocortex[59]. We combined stages 0 (n = 14) and I (n = 78) with II (n = 109), and stage VI (n = 19) with V (n = 290), and made a 4-stage ordinal variable because of few cases in some stages.

### AD pathological diagnosis
A modified Bielschowsky silver stain was used for visualizing diffuse and neuritic plaques, and neurofibrillary tangles in five brain regions. An AD pathological diagnosis was made according to published criteria[2].

### Bulk RNA sequencing
DLPFC was the brain region chosen for further omics studies because of its relatively unique role in cognitive and other higher order human processes and because it exhibits the full range of AD from none to a lot.

Frozen samples of DLPFC were used for extraction of RNA[17,60,61]. In brief, RNA was extracted, concentrated, and sequenced following standard protocols with minor modifications. RNA-seq data were aligned to a human reference genome and transcripts' raw counts were calculated, aggregated at gene levels, and normalized to adjust a sequence bias from GC content and gene length. Finally, appropriate models were applied to remove major technical confounding factors including postmortem interval, sequencing batch, and RNA quality number.

### snRNA-seq
Single nuclei RNA expression profiling was performed on tissues from DLPFC[62]. The tissues were processed in 60 batches, and each batch consisted of 8 participants. In each batch, nuclei suspension of the 8 participants were mixed, and single-nucleus RNA-seq library was prepared using 10x Genomics 3 Gene Expression kit (v3 chemistry). Next, the libraries were sequenced, and unique molecular identifier counting and read mapping were performed using CellRanger v6.0.0 with GENCODE v32 and GRCh38.p13. To get the participant information, the original donors of droplets in each batch were inferred by comparison, that is, matching the sequence variants in RNA reads with ROSMAP Whole Genome Sequencing (WGS) Variant Call Format using the software demuxlet for demultiplexing. For quality control analysis, genotype concordance of RNA and WGS, sex check, duplicated individuals, WGS quality control (QC), and sequencing depth were assessed.

For the cell type annotation step, nuclei were classified into 7 major cell types, and each one of them was analyzed separately. Doublets were removed with the DoubletFinder software, and cells were clustered using the Seurat pipeline.

For downstream analysis, we created pseudo-bulk matrices by summing counts per participant. The genes were filtered by each cell type, keeping genes with count per million reads mapped >1 in 80% of samples. TMM normalization followed by limma *voom* transformation was applied for the final count matrices. Finally, we pulled out the snRNA-seq expression of the *GPER1* gene.

## Targeted proteomic analysis

Selected proteins and peptides were quantified using targeted selection reaction monitoring (SRM) proteomic analysis of DLPFC samples[63,64]. Each participant's tissue (20 mg) was homogenized using a 8 M urea-based denaturation buffer. Following denaturation, protein aliquots were digested using trypsin, and the digests were cleaned using the C18 solid-phase extraction. Tryptic peptide relative abundances were determined using liquid chromatography SRM mass spectrometry after mixing the protein digest with stable isotope-labeled synthetic peptides. SRM data were manually inspected and analyzed using Skyline software[65]. The software calculated the peak area ratios of an endogenous light peptide to its corresponding synthetic peptide, and the peak area ratios were log base 2 transformed and centered at median. The transformed peak area ratios of Aβ peptides (total Aβ: LVFFAEDVGSNK and Aβ38: GAIIGLMVGG) were used in the current study.

## Proteomic profiling of DLPFC

Proteomic profiling was performed on tissue from DLPFC[66,67]. In brief, 100 mg DLPFC tissue was homogenized, sonicated, and centrifuged, and the supernatant was separated. Trypsin and lysyl endopeptidases were used for digesting proteins, which were subsequently labeled using tandem mass tag method. Then, a high-performance liquid chromatography system under high-pH was applied to make fractions to be analyzed by liquid chromatography coupled to mass spectrometry. The spectra were searched against the canonical UniProtKB human proteome database for identification of peptides and proteins based on the assigned spectra. The quality control measures included using global internal standards and regressing out effects of protein batch, MS2 versus MS3 quantitation mode, sex, age at death, postmortem interval, and study (ROS vs. MAP)[68]. From the 8425 quantified proteins, in the current study we examined levels of several proteins involved in cellular processes including GPER1 signaling mechanisms and autophagy.

## Other covariates

Age at death was calculated using birth and death dates. Sex, years of education, and race data were obtained through self-report at the parental cohorts' baseline evaluation. Annually, neuropsychological tests including Mini Mental State Examination were administered to the participants. Participants' scores were reviewed by a neuropsychologist and rated as normal or impaired. A neurologist reviewed the annual clinical data including the neuropsychological tests' scores ratings and determined presence of AD dementia based on established criteria[69]. Medications used by the participants were annually inspected, recorded, and coded using Medi-Span Drug Data Base System[70]. For the current study, we used binary variables to describe whether participants had used estrogen-related medications including estrogens, selective estrogen receptor modulators (such as tamoxifen and raloxifene), or aromatase inhibitors (such as anastrozole and letrozole), in at least 1 year of follow up.

## Statistical analysis

Linear regressions were used to examine the associations of the covariates, including GPER1 RNA expression and Aβ load, with the tau tangles density as the outcome. All the models were controlled for age at death, sex, and education. To examine if the GPER1 modified the association between Aβ and tau tangles, we added to the model an interaction term between GPER1 RNA expression and Aβ load. In subsequent models, we replaced GPER1 with RNA expressions or protein levels of other genes including genes of downstream signaling mechanisms of GPER1 or genes of other G-protein coupled receptors. Finally, we used GPER1 RNA expression level derived from snRNA-seq data. When expression levels of more than one gene were examined, we applied Benjamini and Hochberg false discovery rate (FDR)[71] to adjust for multiple testing. The statistical analyses were supervised by Dr. Lei Yu, PhD at Rush Alzheimer's Disease Center.

## Reporting summary

Further information on research design is available in the Nature Portfolio Reporting Summary linked to this article.

## Data availability

To obtain ROSMAP available resources, qualified investigators should submit an application at www.radc.rush.edu that includes study premises and a short research plan. We deposited bulk-RNA-seq data and protein data on Synapse (https://www.synapse.org/#!Synapse:syn3388564, https://www.synapse.org/#!Synapse:syn17015098). The source data behind the graphs in the paper can be found in Supplementary Data 1.

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

## Acknowledgements

We truly appreciate the ROSMAP participants who diligently devoted their time and efforts to the studies for collection of clinical data, and who consented for brain donation. We also acknowledge staff of Rush Alzheimer's Disease Center for every step of this study including obtaining and analyzing data and data handling. This work was supported by National Institute of Health grants, P30AG010161, P30AG072975, R01AG015819, and R01AG017917, U01AG046152, U01AG061356.

## Author contributions

Shahram Oveisgharan designed and conceptualized study, contributed to the interpretation of the findings, and drafted the manuscript for intellectual content. Lei Yu supervised the statistical analysis and revised the manuscript for intellectual content. Katia de Paiva Lopes contributed in the acquisition and analysis of the data and revised the manuscript for intellectual content. Vladislav A Petyuk contributed in the acquisition and analysis of the data and revised the manuscript for intellectual content. Shinya Tasaki contributed in the acquisition and analysis of the data and revised the manuscript for intellectual content. Ricardo Vialle revised the manuscript for intellectual content. Vilas Menon contributed in the acquisition and analysis of the data and revised the manuscript for intellectual content. Yanling Wang contributed in the acquisition and analysis of the data and revised the manuscript for intellectual content. Philip De Jager contributed in the acquisition and analysis of the data. Julie Schneider contributed in the acquisition and analysis of the data. David A. Bennett contributed in obtaining fund, acquisition and interpretation of the data, and revising the manuscript for intellectual content.

## Competing interests

The authors declare no competing interests.
