## [Peer Review File · Communications Biology]

Reviewers' comments:

Reviewer #1 (Remarks to the Author):

Dear Oveisgharan

The present manuscript approached an important interaction between the main histopathological hallmarks of Alzheimer's disease and the G protein-coupled estrogen receptor (GPER) in postmortem bulk brain tissue. There is a continuing unmet medical need for novel new treatments for patients with Alzheimer's disease especially due to its prevalence in the elderly population. This work brings perspectives for new studies involving the modulation of GPER and the reduction of amyloid- β ($A\beta$) aggregation and tau tangles. However, the manuscript needs major clarification and revision.

Introduction:

The research hypothesis and objectives (last paragraph of the Introduction) are not clearly addressed. Please clarify the main differences between this article and the study already published by your group DOI: 10.1212/WNL.0000000000206833. What is the main contribution of this new study?

Results:

1. Why did you measure $A\beta$ 38 instead of $A\beta$ 42 levels?

Two recent studies indicate that higher cerebrospinal fluid (CSF) levels of $A\beta$ 38 are associated with a smaller decline in the mini-mental state exam (MMSE) score and a reduced risk of conversion to AD (DOI: 10.1212/WNL.0000000000013228 and 10.1002/alz.060772).

2. Regarding the GPER signaling mechanism:

- Is there any direct interaction between the evaluated genes and tau tangles without association with $A\beta$? Since tau tangles are intracellular, greater interaction with these intracellular second messengers are expected.
- Other second messengers may be involved and were not evaluated in this study: Phospholipase C Beta ($PLC\beta$), adenylate cyclase (AC) and protein kinase A (PKA).
- I suggest that the study of downstream signaling pathways of GPER should be evaluated through protein level analysis because you can show activation of pathways by phosphorylation.
- Another pathway that may be involved is autophagy. A recent work has already shown that the induction of estrogen-mediated autophagy can increase the tau clearance (DOI: 10.1016/j.brainres.2022.148079). Markers of the autophagic pathway could be evaluated or discussed in this study.

Discussion

About this sentence:

"Astrocytic GPER1 may be involved in the activation of astrocytes by $A\beta$, which in turn results in the hyperphosphorylation of tau and the formation of tau tangles in neurons through the release of cytokines and other pro-inflammatory factors."

Did you evaluate any cytokines or pro-inflammatory factors? Is there any data in the literature that supports this hypothesis?

Reviewer #2 (Remarks to the Author):

In their study, Oveisgharan and colleagues tested whether GPER1 expression influence the association between amyloid- β ($A\beta$) and tau tangles.

Using post-mortem brains of 1206 individuals (average age at death: 90 years), they show that GPER1 mRNA expression is paralleled with more tau tangles and modifies the association between $A\beta$ load and tau tangles. In particular, GPER1 expression in astrocytes seems to influence the relation between $A\beta$ and tau tangles.

The manuscript is concise and well written. The data presented are intriguing and may be of interest for the community.

However, there are major concerns that the authors should address:

1. The study relies on the hypothesis that extracellular $A\beta$ is somehow triggering tau abnormal phosphorylation and aggregation. However, the progression of $A\beta$ and tau pathology in the brain does not follow the same pattern for both hallmarks: $A\beta$ deposition starts usually in the neocortex, followed by the entorhinal cortex and hippocampus, then the striatum, brain stem and finally the cerebellum. Tau tangles are first observed in the transentorhinal cortex, followed by the entorhinal cortex and hippocampus, then temporal neocortex, association cortices and primary sensory cortices (see also <https://doi.org/10.1016/j.molmed.2022.05.008>)

Are there differences in GPER1 expression in different brain regions? If yes, is it associated with the spreading of $A\beta$ pathology or tau pathology.

2. The term “association” in the title may be misleading because one can interpret that the authors refer to “physical association” or interaction. The title should be modified, given that the authors only show correlation between $A\beta$ pathology, tau pathology and GPER1 expression, and do provide evidence of a cause to effect relationship.

3. The main hypothesis is “that GPER1 might contribute to a pathway that connects extracellular $A\beta$ to intracellular tau tangles”. But soluble $A\beta$ is also found inside the cells, and may therefore influence “directly” tau phosphorylation. Can the authors comment on this point?

4. Why the authors choose to focus on $A\beta$ 38 and not on the toxic $A\beta$ 40 and $A\beta$ 42 peptides? In fact, a recent study showed that having more $A\beta$ 38 protects against AD and slowed cognitive decline (DOI: 10.1212/WNL.00000000000013228).

5. More information should be given in the figure legends, namely the number of individuals and the statistics (especially, because the correlations were not done with the same number of individuals according to the method part).

6. Authors state: “Estrogen level is lower in postmenopausal women compared with men(16)”. However, estradiol can be locally produced in the brain by neurons and astrocytes. This local estradiol synthesis might also explain why authors do not observe a difference between men and women regarding GPER1

expression and its association with A β and tau.

7. The findings regarding GPER1 signaling pathway are not convincing. Indeed, pathways involving JNK, ERK, and Akt rely on activation/inhibition mechanisms by phosphorylation/dephosphorylation, and not likely on gene expression.

8. The conclusion stating that astrocyte GPER1 RNA expression modifies the association between A β and tau tangles may be an overinterpretation of the data. The authors should at least discuss this point in more details in the discussion part.

Minor:

a. The authors should bring more information about GPER1 in the introduction or in the discussion: what is the physiological role in neurons versus astrocytes ? Why this specific estrogen receptor is of specific interest in this study ? What about other estrogen receptors (ER α and ER β) ?

b. Overall, the introduction part is rather short and only scratches the surface of the context of the study.

Reviewer #3 (Remarks to the Author):

In the paper entitled "G-protein coupled estrogen receptor 1 modifies the association of amyloid- β with tau tangles" the authors try to explain how astrocytic GPER1 may be involved in a pathway that links extracellular A β and intracellular tau.

Zhang et al. 2021 reviewed possible synergistic effects between A β and tau on microglial cells and astrocytes (Zhang H, Wei W, Zhao M, Ma L, Jiang X, Pei H, Cao Y, Li H. Interaction between A β and Tau in the Pathogenesis of Alzheimer's Disease. *Int J Biol Sci.* 2021 May 27;17(9):2181-2192. doi: 10.7150/ijbs.57078. PMID: 34239348; PMCID: PMC8241728).

The topic investigated by the author would be of clear interest to the scientific community.

Here are some suggestions and some questions.

Regarding introduction:

Line 55 "and it is 55 unknown what connects these two proteinopathies" _ I would like to recommend this paper: "Nussbaum JM, Schilling S, Cynis H, et al. Prion-like behaviour and tau-dependent cytotoxicity of pyroglutamylated amyloid- β . *Nature.* 2012;485(7400):651-655." even if the authors are focusing on GPER1. There are several hypotheses on the relationship between amyloid- β and tau proteins.

"Unknown connection" looks too simplistic.

Regarding results:

Line 121 "To identify whether the associations obtained by examining bulk RNA expression were due to specific cell types" _ The authors evaluate GPER1 RNA expression levels in different cell types. They evaluate the cell types of neuronal tissue apart from ependymal cells. I would ask if there is a specific reason for this choice.

Regarding online methods:

Line 228 “Sections were immunohistochemically stained using ...” _ The authors refer to sections that were immunohistochemically stained with several antibodies. Unfortunately, there are no figures showing this data. Moreover, line 226 lists multiple brain regions. Are there any differences between the areas reported?

Line 90 “GPER1 RNA expression level was higher in women compared with men (13.88 (SD=0.93) vs. 13.71 (SD=0.84), $p=0.001$), an indirect measure of different estrogen levels in the 2 sexes” is linked to Line 179 Low estrogen levels in postmenopausal women have been suggested to contribute to the vulnerability of women to Alzheimer’s dementia(37, 38) _ The authors focused on estrogenic influence, avoiding commenting on the role of testosterone. Were the participants subjected to estrogen or testosterone therapy during their aging?

REVIEWER 1 COMMENTS:

R1.1 Introduction The research hypothesis and objectives (last paragraph of the Introduction) are not clearly addressed. Please clarify the main differences between this article and the study already published by your group DOI:

10.1212/WNL.000000000206833. What is the main contribution of this new study?

We revised the introduction to more clearly to highlight differences between the current study and our previous work.

Introduction (Lines 78-92):

“To address a possible role for estrogen in more tau in women’s brains¹⁶, in a prior study we examined estrogen receptors, estrogen receptor α (ER1) and β (ER2) and the transmembrane receptor G protein-coupled estrogen receptor (GPER1), in the association with tau tangles and cognitive decline¹⁷. We found that a greater RNA expression of GPER1, not ER1 or ER2, was associated with more tau tangles and faster cognitive decline in women¹⁷. These findings and the transmembrane location of GPER1 were the basis of our hypothesis that GPER1 might contribute to a pathway that connects extracellular A β to intracellular tau tangles. To test this hypothesis, we extended our prior work in several ways. First, we examined an interaction between A β and GPER1 in relation to tau as we expected to observe a stronger association between A β and tau tangles among persons with higher levels of GPER1. Second, we examined both immunohistochemical and proteomic measures of A β in relation to GPER1 and tau tangles. Third, we examined multiple signaling mechanisms of GPER1 to test whether they are also involved in the pathway linking A β with tau tangles. Fourth, we examined single nuclei RNA expression (snRNA-seq) data to identify whether specific cell types’ GPER1 underlie GPER1 relation with A β and tau tangles.”

R1.2 Results.1 Why did you measure A β 38 instead of A β 42 levels?

Two recent studies indicate that higher cerebrospinal fluid (CSF) levels of A β 38 are associated with a smaller decline in the mini-mental state exam (MMSE) score and a reduced risk of conversion to AD (DOI: 10.1212/WNL.000000000013228 and 10.1002/alz.060772).

“The measurement of A β species and a number of other proteins were performed as one multiplexed assay. This assay relied on tryptic digestion of the proteins, following up with quantification of the derived tryptic peptides as the surrogates of the proteins or protein forms. The technical problem with A β 42 is that the derived C-terminal peptide (that is specific to A β 42) is very hydrophobic. It is practically insoluble unless the concentration of acetonitrile or methanol in the sample is higher than 50%. This is incompatible with the current generic sample prep technique used in mass spectrometry proteomics. Thus, the only two peptides reflective of the A β species that we quantified are the generic peptide that maps to the middle of the sequence of any A β species, total A β in the manuscript, and C-terminal fragment of A β 38.

Note, that mass spectrometry measurement of A β 42 is possible [<https://pubmed.ncbi.nlm.nih.gov/23714261/>], however it requires use of an unconventional digestion enzyme – Lys-N. It leaves the N-terminal lysine within the peptide sequence, which helps the solubility. In other words, it takes a dedicated effort to design a mass spectrometry assay specific to A β 42.

R1.3 Results.2.A Regarding the GPER signaling mechanism:

- Is there any direct interaction between the evaluated genes and tau tangles without association with A β ? Since tau tangles are intracellular, greater interaction with these intracellular second messengers are expected.
- Other second messengers may be involved and were not evaluated in this study: Phospholipase C Beta (PLC β), adenylate cyclase (AC) and protein kinase A (PKA).
- I suggest that the study of downstream signaling pathways of GPER should be evaluated through protein level analysis because you can show activation of pathways by phosphorylation.

Thank you for these suggestions. For the suggested genes, we added their RNA expressions to the analyses and examined them altogether with the genes we had already investigated. Then, we examined protein levels of these genes that were quantified in the same dorsolateral prefrontal cortex that RNA expressions were quantified. The protein levels were quantified using tandem mass tag liquid chromatography–mass spectrometry proteomic analysis and were available in 736 out of 1206 participants. We found several of the RNA expressions and protein levels of the GPER1 signaling mechanisms related to tau tangles, but their associations with tau tangles were attenuated in the presence of A β , *GPER1*, and their interactions. We hope these analyses address the reviewer’s comments. The corresponding changes in the results section of the text are provided below.

Results (Lines 144-168):

“Activation of GPER1, which is a transmembrane receptor, results in activation of signaling mechanisms including c-Jun N-terminal kinase (JNK), extracellular signal-regulated kinase (ERK), Akt kinase²⁰, adenylate cyclase, protein kinase A²¹, and Phospholipase C Beta (PLC β)²². We tested whether there was an interaction between RNA expressions and protein levels of these signaling mechanisms and A β in relation to tau tangles.

We first examined the correlations between RNA expression levels of GPER1 and of the genes of downstream signaling mechanisms indicating some coordinated expressions (Table e-6). Then, we examined whether RNA expressions of GPER1 signaling mechanisms modified the associations of A β load with tau tangles. In linear regression models that included terms for RNA expression of each gene separately, A β load, and interaction of the gene with A β , only RNA expression of PLC β 1 had an interaction with A β load (Table e-7).

Next, we examined protein levels of GPER1 signaling mechanisms. GPER1 RNA expression level was correlated with protein levels of some of the signaling mechanisms (Table e-6). When we examined the protein levels of GPER1 signaling mechanisms to find if they modified the associations between A β load with tau tangles, we found that none of the proteins, even PLC β 1, had the modification effect (Table e-7).

Because several of the examined genes were related to tau phosphorylation and tau tangles in prior studies, we examined the GPER1 signaling mechanisms in relation to tau tangles. From the 28 examined RNAs, expression levels of 12 were related to tau tangles. However, only RNA level of one of them remained significantly related to tau tangles when the model also included GPER1, A β load, and their interaction terms (Table e-8). Similarly, from the 26 proteins of GPER1 signaling mechanisms we found 11 to be related to tau tangles with their associations being attenuated in the presence of GPER1, A β , and their interaction (Table e-8). These findings

suggest that some of the examined proteins are related to tau tangles but with a complex relationship that is beyond a simple phosphorylation.”

R1.3 Results.2.B Regarding the GPER signaling mechanism:

- Another pathway that may be involved is autophagy. A recent work has already shown that the induction of estrogen-mediated autophagy can increase the tau clearance (DOI: 10.1016/j.brainres.2022.148079). Markers of the autophagic pathway could be evaluated or discussed in this study.

Thank you for this excellent suggestion. We examined RNA expressions of 22 genes involved in the autophagy pathways, and protein levels of 18 of them were also quantified. We examined them in relation to *GPER1*, A β , and tau tangles, and the corresponding changes of the text are provided below.

Results (Lines 170-184):

“Autophagy is a cellular process characterized by formation of double-membrane vesicles (autophagosomes) around cytoplasmic organelles and molecules, which will coalesce with lysosomes for degradation of engulfed materials and recycle of basic nutritional needs. Autophagy acts during cellular stress including nutrient deprivation and for degrading damaged organelles and proteins²³.

*Prior studies reported involvement of estrogen and estrogen receptors in autophagy²⁴ and autophagy-related tau clearance²⁵. Therefore, we examined RNA expression and protein levels of autophagy genes in relation to *GPER1* RNA level and tau tangles. *GPER1* RNA expression was related to levels of some of the autophagy genes’ RNA expressions and proteins (**Table e-9**). However, none of the 22 autophagy genes’ RNA expression and only 2 of the 18 protein levels modified the association between A β load and tau tangles (**Table e-10**). In addition, while some of the autophagy genes’ RNA expressions and protein levels were associated with tau tangles their associations were attenuated in the presence of *GPER1*, A β , and their interaction (**Table e-11**). These findings suggest that a complex relationship exists between autophagy, A β , tau tangles, and related molecular pathways.”*

R1.4 Discussion. About this sentence:

“Astrocytic *GPER1* may be involved in the activation of astrocytes by A β , which in turn results in the hyperphosphorylation of tau and the formation of tau tangles in neurons through the release of cytokines and other pro-inflammatory factors.”

Did you evaluate any cytokines or pro-inflammatory factors? Is there any data in the literature that supports this hypothesis?

Thank you for raising this point. We examined RNA expressions of 12 astrocytic inflammatory cytokines in relation to *GPER1* RNA expression, A β , and tau tangles and did not find any evidence that supports the above hypothesis. The analyses and corresponding changes in the discussion are provided below.

Results (Lines 21-227):

*“Astrocytes are a key player in neuroinflammation, which is related to Alzheimer’s disease²⁸. Therefore, we examined RNA expressions of 47 inflammatory cytokines using snRNA-seq data and found 12 of them expressed by astrocytes (**Figure e-1**). *GPER1* RNA expression had a null to weak correlation with the RNA expressions of the 12 astrocytic cytokines (**Table e-15**). However, RNA expressions of none of the astrocytic cytokines modified the association between*

Aβ and tau (Table e-16). Moreover, while RNA expressions of 4 of the cytokines were related to tau tangles, their associations were attenuated and not significant after inclusion of Aβ, GPER1 RNA expression, and their interaction (Table e-17). The above analyses indicated that the pathway connecting GPER1 and Aβ with tau does not involve astrocytic cytokines.”

Discussion (Lines 252-259):

“Our finding that astrocytic GPER1 modified the association between Aβ and tau tangles is an indirect evidence that GPER1 may be a link between Aβ and tau. Astrocytes were a key mediator in Aβ-induced tau phosphorylation in neuronal cultures³⁹, and in linking participant’s Aβ burden to plasma phosphorylated tau level⁴⁰. Our analysis did not support release of inflammatory cytokines by activated astrocytes as a mechanism linking Aβ and tau tangles through astrocytic GPER1. One possible mechanism is GPER1-mediated internalizing of Aβ in the astrocytes^{41,42} that triggers formation of astrocytic tau aggregates⁴³, which act as seeds for formation of neuronal tau tangles.”

REVIEWER 2 COMMENTS:

R2.1 The study relies on the hypothesis that extracellular Aβ is somehow triggering tau abnormal phosphorylation and aggregation. However, the progression of Aβ and tau pathology in the brain does not follow the same pattern for both hallmarks: Aβ deposition starts usually in the neocortex, followed by the entorhinal cortex and hippocampus, then the striatum, brain stem and finally the cerebellum. Tau tangles are first observed in the transentorhinal cortex, followed by the followed by the entorhinal cortex and hippocampus, then temporal neocortex, association cortices and primary sensory cortices (see also <https://doi.org/10.1016/j.molmed.2022.05.008>)

Are there differences in GPER1 expression in different brain regions? If yes, is it associated with the spreading of Aβ pathology or tau pathology.

In subsets of participants data of *GPER1* RNA expression level in the anterior caudate (n=687) and the posterior cingulate cortex (n=633) were available. Although *GPER1* levels in the 3 brain regions were different (Table e-2), they were moderately correlated (all 3 $\rho > 0.5$, Table e-2). Like *GPER1* level in DLPFC, *GPER1* RNA levels in the posterior cingulate cortex was related to higher levels of tau tangles and modified the association of Aβ load with tau tangles, but not *GPER1* in the anterior caudate (Table e-3). Then, we determined stages of Aβ and tau tangle distribution across the brain using a modified 4-level Thal (lines 345-349) and Braak stages (lines 354-361), respectively. Replacing the global measures of Aβ load and tau tangle density with the stages of Aβ and tau tangle distribution did not change our main findings. Please find below the revised results and discussion sections to include the above findings.

Results (Lines 130-142):

“In subsets of participants GPER1 RNA expression levels were also determined in the posterior cingulate cortex (n=633) and the anterior caudate (n=687). Although GPER1 levels in the 3 brain regions were different (Table e-2), the levels were moderately correlated (all 3 $\rho > 0.5$, Table e-2). Like GPER1 level in DLPFC, GPER1 RNA level in the posterior cingulate cortex was related to higher levels of tau tangles and modified the association of Aβ load with tau tangles, but not GPER1 in the anterior caudate (Table e-3). When we replaced global measures of Aβ load and tau tangle density with the stages of Aβ and tau tangle distribution across the

brain, the main findings persisted. A higher GPER1 RNA level in DLPFC was associated with more advanced stages of tau tangles (Table e-4). In addition, GPER1 RNA level in both DLPFC and the posterior cingulate cortex had a borderline interaction with the stage of A β in relation to tau tangle stages. However, GPER1 RNA level in the anterior caudate did not have these associations (Table e-4). GPER1 RNA level in neither of the brain regions was associated with the stage of A β or A β load (Table e-5)."

Discussion (Lines 278-285):

"GPER1 RNA expression level in the anterior caudate was not related to tau tangles and did not modify the associations of A β with tau tangles, in contrast to GPER1 RNA level in DLPFC and the posterior cingulate cortex. Activation of neuronal GPER1 had different effects dependent on the neuronal region⁵², which may underlie heterogenous associations of GPER1 with tau tangles in the current study. However, we did not measure A β in the caudate, which might be lower than A β level in DLPFC considering distribution pattern of A β that begins from neocortex and spreads down to the basal ganglia including caudate nucleus and may underlie lack of association between GPER1 RNA level in the anterior caudate and tau tangles."

R2.2 The term "association" in the title may be misleading because one can interpret that the authors refer to "physical association" or interaction. The title should be modified, given that the authors only show correlation between A β pathology, tau pathology and GPER1 expression, and do provide evidence of a cause to effect relationship.

We revised the title to address the reviewer's concern. The revised title is: "G-protein coupled estrogen receptor 1, amyloid- β , and tau tangles in older adults".

R2.3 The main hypothesis is "that GPER1 might contribute to a pathway that connects extracellular A β to intracellular tau tangles". But soluble A β is also found inside the cells, and may therefore influence "directly" tau phosphorylation. Can the authors comment on this point?

We agree with the reviewer. Please find below changes in the discussion that addresses this point.

Discussion (Lines 257-262):

"One possible mechanism is GPER1-mediated internalizing of A β in the astrocytes^{41,42} that triggers formation of astrocytic tau aggregates⁴³, which act as seeds for formation of neuronal tau tangles. This hypothesis is supported by prior studies that showed presence of A β oligomers inside the cells and induction of tau phosphorylation after intraventricular injection of A β oligomers in monkeys⁴⁴ or treating cell cultures by solutions containing soluble A β oligomers⁴⁵."

R2.4 Why the authors choose to focus on A β 38 and not on the toxic A β 40 and A β 42 peptides? In fact, a recent study showed that having more A β 38 protects against AD and slowed cognitive decline (DOI: 10.1212/WNL.0000000000013228).

Please see **R1.2 Results.1** above.

R2.5 More information should be given in the figure legends, namely the number of individuals and the statistics (especially, because the correlations were not done with the same number of individuals according to the method part).

We added legends to the figures and described the examined models and their sample sizes.

R2.6 Authors state: “Estrogen level is lower in postmenopausal women compared with men(16)”. However, estradiol can be locally produced in the brain by neurons and astrocytes. This local estradiol synthesis might also explain why authors do not observe a difference between men and women regarding GPER1 expression and its association with A β and tau.

We agree. Please see below a section of the revised discussion that highlights this point.

Discussion (Lines 274-277):

“However, in our study we did not measure estrogen level in DLPFC tissue. Because local estrogen is synthesized in the brain⁵¹ that may offset low levels of estrogen in the circulation, further studies are needed to explicate the relationship between estrogen, GPER1, A β , and tau in women and men.”

R2.7 The findings regarding GPER1 signaling pathway are not convincing. Indeed, pathways involving JNK, ERK, and Akt rely on activation/inhibition mechanisms by phosphorylation/dephosphorylation, and not likely on gene expression.

We agree with the reviewer. Although we added examination of the protein levels of JNK, ERK, and Akt to partially address this concern (see **R1.3** above), we did not examine their activation level, which is added to the limitation section of the manuscript.

Discussion (Lines 302-304):

“We tested RNA and protein levels of GPER1 signaling mechanisms including JNK, ERK, and Akt kinases instead of their activation level.”

R2.8 The conclusion stating that astrocyte GPER1 RNA expression modifies the association between A β and tau tangles may be an overinterpretation of the data. The authors should at least discuss this point in more details in the discussion part.

Upon reflection, we agree with the reviewer and revised the limitation section of the discussion to more clearly indicate the above concern.

Discussion (Lines 291-296):

“However, several limitations must be noted. All the findings were derived from an observational study, and our main finding that astrocytic GPER1 modifies the association between A β and tau does not imply a cause and effect finding unless confirmed by future experimental studies. Although many molecular pathways were examined, still some unmeasured confounders rather than GPER1 may underlie the modification of the relation between A β and tau.”

R2.9 Minor.a The authors should bring more information about GPER1 in the introduction or in the discussion: what is the physiological role in neurons versus astrocytes ? Why this specific estrogen receptor is of specific interest in this study ? What about other estrogen receptors (ER α and ER β) ?

Please find sections of the introduction and discussion below that are revised to address this comment.

Introduction (Lines 76-84):

“Women have higher levels of tau tangles compared with men^{12,13}, and low estrogen levels in postmenopausal women have been suggested to contribute to vulnerability of women to AD dementia^{14,15}. To address a possible role for estrogen in more tau in women’s brains¹⁶, in a prior study we examined estrogen receptors, estrogen receptor α (ER1) and β (ER2) and the transmembrane receptor G protein-coupled estrogen receptor (GPER1), in the association with tau tangles and cognitive decline¹⁷. We found that a greater RNA expression of GPER1, not ER1 or ER2, was associated with more tau tangles and faster cognitive decline in women¹⁷. These findings and the transmembrane location of GPER1 were the basis of our hypothesis that GPER1 might contribute to a pathway that connects extracellular A β to intracellular tau tangles.”

Discussion (Lines 263-270):

“ER1 and ER2 are essentially nuclear estrogen receptors that after activation by estrogen bind transcription factors and regulate transcription of a variety of genes involved in different cellular processes. Besides the genomic effect that occurs in the time frame of hours to days, estrogen has an instant non-genomic effect that starts in the cells within minutes of estrogen exposure, contributes to estrogen-induced neuroprotection⁴⁶ and cognitive function⁴⁷, and is mediated by estrogen surface receptors including GPER1⁴⁸. In fact, prior experimental studies reported that activation of GPER1 by estrogen has cell-type specific consequences as it promotes cell survival in neurons and causes apoptosis in astrocytes⁴⁹.”

R2.10 Minor.b Overall, the introduction part is rather short and only scratches the surface of the context of the study.

We revised the entire introduction (Lines 52-92).

REVIEWER 3 COMMENTS:

R3.1 Zhang et al. 2021 reviewed possible synergistic effects between A β and tau on microglial cells and astrocytes (Zhang H, Wei W, Zhao M, Ma L, Jiang X, Pei H, Cao Y, Li H. Interaction between A β and Tau in the Pathogenesis of Alzheimer's Disease. Int J Biol Sci. 2021 May 27;17(9):2181-2192. doi: 10.7150/ijbs.57078. PMID: 34239348; PMCID: PMC8241728). The topic investigated by the author would be of clear interest to the scientific community.

Thank you for raising the point. We revised the introduction (lines 62-74) and discussed in more details the synergism between A β and tau and cited the mentioned paper (Ref. 3).

R3.2 Regarding introduction: Line 55 “and it is 55 unknown what connects these two proteinopathies” _ I would like to recommend this paper: “Nussbaum JM, Schilling S, Cynis H, et al. Prion-like behaviour and tau-dependent cytotoxicity of pyroglutamylated amyloid- β . Nature. 2012;485(7400):651-655.” even if the authors are focusing on GPER1. There are several hypotheses on the relationship between amyloid- β and tau proteins. "Unknown connection" looks too simplistic.

Thank you for the reference. The mentioned paper is now cited in the revised introduction (Ref. 10).

R3.3 “To identify whether the associations obtained by examining bulk RNA expression were due to specific cell types” _ The authors evaluate GPER1 RNA expression levels in different cell types. They evaluate the cell types of neuronal tissue apart from ependymal cells. I would ask if there is a specific reason for this choice.

Quantification of single nuclei RNA expression was performed on samples from dorsolateral prefrontal cortex where ependymal cells are not present. Ependymal cells delineate ventricles and could be detected in tissues around the ventricles.

R3.4 Line 228 “Sections were immunohistochemically stained using ...” _ The authors refer to sections that were immunohistochemically stained with several antibodies. Unfortunately, there are no figures showing this data.

We added figures to the supplementary file to illustrate sections immunohistochemically stained with antibodies against A β and phosphorylated tau (**Figure e-2**).

R3.5 Moreover, line 226 lists multiple brain regions. Are there any differences between the areas reported?

The A β load in the 8 regions were highly correlated as the bivariate Spearman correlation coefficients were between 0.81 – 0.95, supporting summarizing them with an average global measure. However, averages of the 8 regional A β measures were different (F=81.2, p<0.001), with the highest average in the middle frontal cortex, where we had measured *GPER1* RNA expression, and the lowest in the hippocampus. Moreover, please see **R2.1** above where we examined the association between *GPER1* RNA expression in 3 brain regions with A β and tau tangles.

We added the averages of the 8 regional A β measures and their correlations to **Table e-18**. Moreover, the methods section was revised, which is provided below.

Methods (Lines 343-345):

“The averages of the percent areas were calculated to yield regional A β loads, which were subsequently averaged to yield brain A β load because of being highly correlated (Table e-18)^{16,30}.”

R3.6 Line 90 “GPER1 RNA expression level was higher in women compared with men (13.88 (SD=0.93) vs. 13.71 (SD=0.84), p=0.001), an indirect measure of different estrogen levels in the 2 sexes” is linked to

Line 179 Low estrogen levels in postmenopausal women have been suggested to contribute to the vulnerability of women to Alzheimer’s dementia(37, 38) _ The authors focused on estrogenic influence, avoiding commenting on the role of testosterone. Were the participants subjected to estrogen or testosterone therapy during their aging?

Between 23 to 91 participants (2% to 8%) had used different classes of estrogen-related medications (**Table 1**). However, our main findings did not change after controlling for estrogen-related medications, as described below.

Although only 5 participants had used androgen-related medications, we agree with the reviewer that other sex-related factors should also be investigated. We added this to the limitation section of the discussion and is provided below.

Methods (Lines 427-431):

“Medications used by the participants were annually inspected, recorded, and coded using Medi-Span Drug Data Base System⁶⁸. For the current study, we used binary variables to describe whether participants had used estrogen-related medications including estrogens, selective estrogen receptor modulators (such as tamoxifen and raloxifene), or aromatase inhibitors (such as anastrozole and letrozole), in at least 1 year of follow up.”

Results (Lines 127-129):

“Furthermore, controlling the main models for estrogen-related medications did not change the modification of the association between A β and tau tangles by GPER1 (Table e-1).”

Discussion (Lines 304-307):

“The current study examined GPER1 as a transmembrane estrogen receptor with a potential to link extracellular A β with intracellular tau tangles. Future studies should examine other receptors of sex and steroid hormones that are located intracellularly and may be involved in AD pathophysiology.”

REVIEWERS' COMMENTS:

Reviewer #1 (Remarks to the Author):

All Raised Questions have been appropriately answered. I have just two suggestions:

1. In the discussion you could address the limitations of evaluating A β 42 levels and new studies that link A β 38 levels and Alzheimer's disease.
2. Authors state about GPER1 signaling results: "These findings suggest that some of the examined proteins are related to tau tangles but with a complex relationship that is beyond a simple phosphorylation." (Line 166-168). However, in this study you did not evaluate the phosphorylation and activation of proteins downstream of the GPER1 pathway. Could you explain how you came to this conclusion? You could talk about it in the discussion.

Reviewer #2 (Remarks to the Author):

I would like to thank the authors for incorporating all the amendments suggested for the manuscript. I believe these changes have significantly enhanced the overall quality of the document. Consequently, I have no further comments to add. In my opinion, the manuscript is now suitable for publication.

Reviewer #3 (Remarks to the Author):

In the paper entitled "G-protein coupled estrogen receptor 1 modifies the association of amyloid- β with tau tangles" the authors try to explain how astrocytic GPER1 may be involved in a pathway that links extracellular A β and intracellular tau.

The authors rewrote the paper with better data than the previous version. Even the aim of the work seems clearer than before, and the discussion is open to future evaluations.

REVIEWER 1 COMMENTS:

R1.1 In the discussion you could address the limitations of evaluating A β 42 levels and new studies that link A β 38 levels and Alzheimer's disease.

Thank you for the suggestion. We added a paragraph to the discussion to address the reviewer's point, which is provided below.

Discussion (Lines 286-294):

“Besides using immunohistochemical methods to measure A β load, we also measured level of A β through targeted proteomic analysis, which relies on the identification of unique protein sequences. The only two A β species quantified were the generic peptide that maps to the middle of the sequence of any A β species, identified as total A β protein, and a C-terminal fragment of A β 38. Notably, we were not able to quantify A β 42 because the derived C-terminal peptide (that is specific to A β 42) is very hydrophobic that made it incompatible with the current generic sample preparation technique used in mass spectrometry proteomics. However, recent studies find that A β 38 is elevated in both familial and sporadic forms of AD⁵⁴, which support using A β 38 in our study.”

R1.2 Authors state about GPER1 signaling results: “These findings suggest that some of the examined proteins are related to tau tangles but with a complex relationship that is beyond a simple phosphorylation.” (Line 166-168). However, in this study you did not evaluate the phosphorylation and activation of proteins downstream of the GPER1 pathway. Could you explain how you came to this conclusion? You could talk about it in the discussion.

Thank you for raising the concern. We revised the sentence and removed the reference to phosphorylation. We also added a paragraph to explain the finding that the protein levels of GPER1 signaling mechanisms were related to tau tangles, but their associations were attenuated in the presence of GPER1, A β , and their interaction.

Results (Lines 166-168):

“These findings suggest that some of the examined proteins are not directly related to tau tangles, but rather are associated by a complex relationship that involves GPER1, A β , and their interaction.”

Discussion (Lines 295-306):

“Approximately all GPER1 signaling mechanisms proteins were kinases that act through phosphorylating their target. As abnormal tauopathy is initiated by hyperphosphorylation of tau, one hypothesis was that activation of GPER1 is transduced intracellularly by activation of these kinases that ends in hyperphosphorylation of tau and production of tau tangles. This hypothesis was supported by our finding that the protein levels of several of these kinases were related to tau tangle density. However, the relationship between the kinase levels and tau tangle density was attenuated in the presence of A β , GPER1, and their interaction in the model, which is not consistent with the kinases being mediators of the activation of GPER1 by A β . In statistical mediation, the association between the mediator and the outcome does not attenuate in the presence of the upstream cause. Therefore, we concluded that a complex relationship exists between A β , GPER1 and their signaling mechanisms, and tau tangles, which requires further studies for clarification.”

REVIEWER 2 COMMENTS:

I would like to thank the authors for incorporating all the amendments suggested for the manuscript. I believe these changes have significantly enhanced the overall quality of the document. Consequently, I have no further comments to add. In my opinion, the manuscript is now suitable for publication.

Thanks for your review that helped in improving the manuscript.

REVIEWER 3 COMMENTS:

In the paper entitled “G-protein coupled estrogen receptor 1 modifies the association of amyloid- β with tau tangles” the authors try to explain how astrocytic GPER1 may be involved in a pathway that links extracellular A β and intracellular tau.

The authors rewrote the paper with better data than the previous version. Even the aim of the work seems clearer than before, and the discussion is open to future evaluations.

Thanks for your review that helped in improving the manuscript.